# Effect of Knife Use and Overlapping Gloves on Finger Temperature of Poultry Slaughterhouse Workers

**DOI:** 10.3390/ijerph21101314

**Published:** 2024-10-01

**Authors:** Adriana Seára Tirloni, Diogo Cunha dos Reis, Antônio Renato Pereira Moro

**Affiliations:** 1Labor Prosecution Service, PRT12, Florianópolis 88025-255, SC, Brazil; adri@tirloni.com.br; 2Biomechanics Laboratory, CDS, Federal University of Santa Catarina, Florianópolis 88040-900, SC, Brazil; renato.moro@ufsc.br

**Keywords:** thermography, skin temperature, gloves, personal protective equipment, slaughterhouse, risk assessment, ergonomics

## Abstract

Brazilian poultry slaughterhouses employ many workers, consequently exposing them to various ergonomic risks. This study aimed to analyze the effects of knife use and overlapping gloves on the finger temperatures of poultry slaughterhouse workers. Employees (*n* = 571) from seven Brazilian poultry slaughterhouses participated in this cross-sectional study. A Flir^®^ T450SC infrared camera was used to record thermographic images of the workers’ hands. The workers were interviewed about work organization, cold thermal sensations, and the perception of upper-limb musculoskeletal discomfort. Dependent and independent sample *t*-tests and binary logistic regression models were applied. The results proved that the workers wore up to five overlapping gloves and had at least one finger with temperatures of ≤15 °C (46.6%) or ≤24 °C (98.1%). Workers that used a knife and wore a chainmail (CM) glove on their non-dominant hand had average finger temperatures significantly colder on the palmar surface than the anti-cut (AC) glove group (*p* = 0.029). The chance of one worker who wore a CM glove to have finger temperatures of ≤15 °C was 2.26 times greater than a worker who wore an AC glove. Those who wore an AC glove and those wearing a CM glove presented average overall finger temperatures significantly lower on the non-dominant hand (products) than the dominant hand (knife) (*p* < 0.001).

## 1. Introduction

Brazilian poultry slaughterhouses employ many workers, as the last record reported that Brazil was the leader in poultry meat exports and the second-largest producer in the world [1]. Consequently, thousands of these workers are exposed to ergonomic risks. OSHA [2] stated that these risks may lead to the development of musculoskeletal disorders (MSDs) in poultry processing, which are as follows: repetition, forceful exertion, awkward and static postures, vibration, and cold temperatures.

Studies have shown that the artificially cold environment (cutting room) of Brazilian poultry slaughterhouses does not exceed 12 °C [3,4]. Therefore, regulatory norm 36 (NR-36) regarding slaughterhouses recommends that the personal protective equipment (PPE) and clothing worn must be compatible with the temperature of the work environment, in order to promote thermal comfort [5]. It should be highlighted that when cold is combined with exposure to other risk factors, it can increase the hazard of developing upper-limb work-related musculoskeletal disorders (UL-WMSDs) [2].

Several types of gloves are used as PPE to protect the hands against abrasive and scoring agents; sharp objects and piercing; electric shocks; thermal and biological agents; chemicals; vibrations; humidity from operations using water; and ionizing radiation [6]. Investigations into the effects of gloves have been performed. During a screw-driving task, wearing gloves significantly increased muscle activity, pinch strength, and discomfort, but reduced dexterity and touch sensitivity [7]. Fry et al. [8] verified that, for surgeons, double-gloving does not have a substantial impact on manual dexterity or tactile sensitivity when compared with no gloves or single-gloving. In slaughterhouses, tasks may require the use of more than one glove type on the same hand, protecting against cold, humidity, and cuts [9]. One paper proved that these workers wore up to five overlapping gloves, with an average of three gloves on each hand, yet most of them still felt cold in their hands [3]. Additionally, using thermography images confirmed that most poultry workers had mean finger temperatures of ≤15 °C (high physiological strain) [3,4]. 

Vogt [10] points out that, when a worker is subjected to cold stress, the effects (in seconds) are a cutaneous response (cooling) and discomfort. Lehmuskallio et al. [11] stated that a discomfort sensation (cold) of the skin occurs when the skin temperature is <25 °C. According to Vogt [10], the importance of studying finger temperatures and the use of gloves is due to the fact that cooling muscle tissue reduces blood flow and slows down some neural processes, such as the transmission of nerve signals and synaptic functions.

In addition to the aforementioned risks to which slaughterhouse workers’ hands are exposed (cold and glove use), there is also a risk regarding the use of hand tools. In total, 73.5% of the types of products leading the exports of Brazilian poultry slaughterhouses include cuts [1]. According to OSHA [2], workers undergo exertion to maintain control of equipment or tools, and conforming to Arvidsson et al. [12], cutting tasks require significant efforts of flexion and extension of the wrist. Studies have confirmed the relationship between tool use and the development of UL-WMSDs [13,14,15], as well as the handle diameter and hand grip strength [16]. The NR-36 manual recommends that, in slaughterhouse tasks that require workers to wear overlapping gloves, the tool handle diameter used should be considered as their hand size becomes larger, increasing the effort needed when performing the task [9].

Although there are studies on the finger temperatures of poultry slaughterhouse workers and their relationship with tool use [3,4], no studies were found to verify the sequence of the gloves worn on the same hand, as well as their effects on thermal protection. The research question is “What sequence of overlapping gloves promotes greater thermal protection for the fingers of poultry slaughterhouse workers when using a knife or not?”. Thus, this study aimed to analyze the effects of knife use and overlapping gloves on the finger temperatures of poultry slaughterhouse workers. 

## 2. Materials and Methods

Seven Brazilian poultry slaughterhouses participated in this cross-sectional study, developed with ethical approval from the Federal University of Santa Catarina, protocol nº 2098/2011. Organizational work characteristics were provided by the health and safety teams of the slaughterhouses and are described in Table 1.

The research was limited to analyzing only those employees who worked in an artificially cold environment (≈12 °C).

### 2.1. Participants

The selection of poultry slaughterhouses was intentional, but the selection of workers was random. In the artificially cold environment, the researcher selected the first worker at the table or line, skipped one and called the next, and so on. To eliminate the possibility of skin temperature alterations, the following exclusion criteria were adopted: does not smoke [17], is not sleep deprived prior to the assessment [18], and no alcoholic beverages consumed 12 h preceding the data collection [19]. 

Thereby, the initial sample was 600 workers, but only those who had thermographic images of both hand surfaces (palmar and dorsal) captured were included in the study; thus, the final sample was 571 participants (Table 1). Of these, 374 workers used a knife and 197 did not. All invited employees agreed to participate in the study.

### 2.2. Instruments

An infrared portable Flir^®^ T450SC camera (Flir Systems, Wilsonville, OR, USA) with an 18 mm lens was used to record the thermographic images. The Flir^®^ Tools software version 6.4.18039.1003 was used to analyze the images (palm and dorsum of the hands).

The selected workers were interviewed about identification data (age), work organization (time working at the slaughterhouse), thermal sensations of cold (if the worker felt cold in their hands and at what intensity), and perceptions of upper-limb musculoskeletal discomfort in the last 12 months. The following symptoms were considered as bodily discomfort: pain, fatigue, shocks, cracks, numbness, tingling, weight, strength loss, and movement limitation in the upper limbs [20]. To identify thermal sensations, a numerical scale was used to evaluate the cold feeling in the hands, where 0 indicated feeling neutral and −1, −2, and −3 indicated feeling slightly cold to very cold [21].

### 2.3. Gloves

The workers used personal protective equipment (PPE) for their hands, alongside clothing, aprons, socks, and boots provided by the slaughterhouses with a Certificate of Approval from the Brazilian Ministry of Labor. 

Nine types of gloves used by workers were found, classified according to their functionality as follows: waterproof, cut protection, and thermal (Figure 1). All the surveyed gloves were produced based on international standards.

The main function of the anti-cut gloves is mechanical protection, but they are also considered to provide thermal protection (secondary function) (Figure 1). Despite this, in the present study, anti-cut gloves were classified as having mechanical resistance.

### 2.4. Procedures of Data Collection

Data collection was performed during two slaughterhouse work shifts at or near workstations. Following the recommendations proposed by ISO 11079 [22], the data collection was started only when the workers were in the artificially cold room performing their activity at the workstation for at least 15 min. 

The worker was instructed to stop working, go to the collection site, and remove their gloves. The number and type of gloves that the worker was using at the moment of data collection were verified and annotated (from outer to inner) by the researcher when the employee took them off. After that, the participants positioned themselves for the collection of the thermographic images and then were interviewed.

The camera was placed approximately 1 m away from the participant and 0.9 m above the floor. Information about the temperature (≈12 °C) and air humidity (50%) of the room, in addition to the emissivity of the human body (0.98), was obtained with the Flir^®^ software for later analysis of the images.

Two thermographic images of the palmar and dorsal surfaces of the hands were collected from each worker. The identification of the coldest area of each finger was performed using the software’s ellipse tool (at least 7 pixels in diameter), avoiding the edges of the fingers (Figure 2). The average finger temperatures were identified by the software for subsequent statistical data analysis. 

### 2.5. Study Groups

Initially, the participants were separated into the two following groups: GK—the group of workers who used a knife (*n* = 374 workers) and GWK—the group of workers who did not use a knife (*n* = 197 workers). Subgroups were formed according to the sequence of gloves worn by the workers to compare the different sets of gloves used only on the non-dominant hand (COMP1) and compare between the sets of gloves used on the non-dominant and dominant hands (COMP2) (Figure 3).

### 2.6. Statistics

The statistical analysis was performed using IBM SPSS Statistics, version 21.0 (IBM Corp., Armonk, NY, USA). With the objective of grouping the data in relation to the workers’ manual dominance, the left-handers were identified, and the data on finger temperatures and the number and types of gloves were inserted into the data sheet.

To compare the glove sets worn on the non-dominant hand (COMP1), both for GK.CM and GK.AC, sample homogeneities were analyzed and the Student’s independent *t*-test was applied. Paired sample *t*-tests were used to compare the glove sets worn between the non-dominant and dominant hands (COMP2). For this test, workers who wore different glove sets were excluded, as they formed a group with a very small sample.

ISO 11079 “Ergonomics of the thermal environment” was used as a reference to classify the finger temperatures based on the physiological criteria described [22]. This norm establishes that finger temperatures of ≤15 °C are considered to impart a high physiological strain, while those of ≤24 °C imply a low strain. 

The difference between the left and the right hand was calculated considering predetermined regions, and the criterion adopted for the acceptable limits of thermal asymmetry between both hands was <1 °C [23]. To compare the delta of the finger temperatures between the non-dominant and dominant hands with the different glove sets and knife use, the Student’s independent *t*-test and one-way analysis of variance (Tukey test) were used.

Binary logistic regression models were used to assess the associations between the types of cut protection gloves (chainmail or anti-cut gloves) on the non-dominant hand and the independent variables, which were as follows: thermal sensations of cold in the hand, perceptions of discomfort in the upper limbs (overall), and finger temperatures (≤15 °C or >15 °C). The odds ratios and confidence intervals (95% CI) for the types of cut protection gloves were estimated for crude and adjusted analyses. First, the crude model was applied between the types of cut protection gloves and the independent variables separately. For the variables to be included in the adjusted model, they had to present *p* < 0.20 in the crude model. A statistical significance level of *p* ≤ 0.05 was adopted for all tests.

## 3. Results

The employees had worked for the companies from 3 months to 29 years (3.6 ± 4.6 years), performing their tasks in the cutting room (91.2%) and other sectors (8.8%). Most of the workers were female (70.4%), right-handed (95.1%), and used a knife (65.5%). Almost half of the workers felt cold in at least one hand (46.1%) and had at least one finger with a temperature of ≤15 °C (46.6%). However, most workers had fingers with temperatures of ≤24 °C (98.1%). Additionally, 49.9% felt musculoskeletal discomfort, with the shoulder (32.9%) and wrist (12.6%) regions being the most affected (Table 2).

This study showed that the workers wore from one to five overlapping gloves to protect against humidity, cuts, and cold. In general, the workers used more than a waterproof glove (nitrile and/or plastic) on the left (64.3%) and right hands (68.1%). When only analyzing the group that utilized tools, most workers that used a knife wore two or more overlapping waterproof gloves on their non-dominant (70.6%) and dominant hands (76.5%), however, this had a lower prevalence for both hands in the group that did not use this tool (52.3%).

Wearing three overlapping gloves was the most common condition overall, and for workers who used or did not use a knife, this regarded both on their non-dominant (53.1%) and dominant hands (48.3%) (Table 3). It is noteworthy that 49.2% of the workers who used a knife felt cold in their non-dominant hand.

Most workers who used a knife wore a chainmail glove on their non-dominant hand (60.7%) for cut protection, and of these, 37% wore an anti-cut glove on their dominant hand. Unexpectedly, two workers who wore a chainmail glove and did not use a knife wore this glove type to pull the chicken’s skin in order to increase friction. Of the seven slaughterhouses surveyed, five adopted chainmail gloves and two adopted anti-cut gloves for the non-dominant hands. It was found that 42 (11.2%) and 60 (30%) workers of those who used (*n* = 374) and did not use a knife (*n* = 197), respectively, wore anti-cut gloves on both hands. Most of the GK workers did not wear the specific thermal protection glove (49.2%) on their non-dominant hands, and for GWK, 62.4% did not wear this type of glove on their dominant hands (Table 3), which remained in contact with the products. In the group that used a knife, it was found that six workers wore a thermal glove below the anti-cut glove on their dominant hand (1.6%).

In the GK.CM, the GK.CM1 presented average finger temperatures significantly lower than those of the GK.CM2, because GK.CM1 workers did not use a thermal glove below the chainmail glove (*p* < 0.001), even though they wore more than one waterproof glove (Table 4). Furthermore, 70.7% of GK.CM1 workers had average finger temperatures of ≤15 °C. However, for GK.AC, there was no significant difference between the GK.AC1 and the GK.AC2 (*p* > 0.05), despite the fact that they used more than one waterproof glove with different sequences from those worn by GK.AC1 (Table 4).

As for the non-dominant hand, 51.1% of workers who wore a CM glove and 32.7% who wore an AC glove had at least one finger with an average temperature of ≤15 °C. Moreover, when comparing the average finger temperatures between the two groups (GK.CM and GK.AC), it was evident that the hand that wore the CM glove was significantly colder on the palmar surface than GK.AC (*p* = 0.029), unlike the dorsal surface (*p* = 0.064). In the individual analysis, fingers 2 and 3 showed temperatures significantly lower with the CM glove than the AC glove on the palmar surface (*p* = 0.016 and *p* = 0.005, respectively) (Table 4).

The discomfort prevalence of the upper limbs in the group using a knife was 54.3% (Table 2). When performing the binary regression in relation to the types of cut protection gloves, the crude assessment identified that there were no associations between glove types and thermal sensations of cold in the non-dominant hands (*p* = 0.722). However, there was an association between discomfort and average finger temperatures of ≤15 °C in the non-dominant hand (*p* ≤ 0.20). These two variables were evaluated in the adjusted model and showed that the chance of one worker who wore a CM glove to have average finger temperatures of ≤15 °C was 2.26 times greater than a worker who wore an AC glove (OR = 2.26, 95% CI 1.46; 3.51), controlling the discomfort.

In all comparisons of the COMP2.GK.CM between the non-dominant and dominant hand surfaces, the overall temperatures of all fingers were significantly lower on the hand that wore a CM glove and manipulated the product (*p* < 0.001), regardless of wearing a thermal glove under the CM glove (Table 5). Most of these workers had average finger temperatures that were lower on the non-dominant hand than on the dominant hand on both the palmar (74.3%) and dorsal surfaces (81.2%). In general, most workers who wore a CM glove on the non-dominant hand had one finger with an average temperature of ≤15 °C (50.9%) and wore three (48.2%) or four gloves (45.4%). Contrarily, regarding the dominant hand, 30.7% of workers had average finger temperatures of ≤15 °C, and the majority wore three gloves (54.4%).

Of the 218 workers who wore CM gloves, 60.6% and 62.8% of workers had a temperature delta between finger temperatures (dominant–non-dominant) of >1 °C on the palmar and dorsal surfaces, respectively. The fingers obtained a minimum delta of −7.9 °C and a maximum of 12.5 °C. When comparing the deltas of finger temperatures between the COMP2.GK.CM groups, it was found that there was no significant difference between the groups (*p* > 0.05).

Table 6 shows that the temperatures of all fingers were significantly lower on the hand that wore an AC glove and manipulated the product (*p* < 0.001) (non-dominant hand) than on the opposite hand. This corroborates that, in general, most of these workers had lower finger temperatures on the non-dominant hand than the dominant hand on the palm (80.3%) and dorsum (83%).

Of the 147 workers who wore AC gloves, 59.2% and 63.9% of workers had a temperature delta between the fingers of >1 °C on the palmar and dorsal surfaces, respectively. The fingers achieved a minimum delta of −15.0 °C and a maximum of 14.8 °C.

Comparing the temperature deltas between the COMP2.GK.AC groups, a significantly lower palm temperature delta was found in the COMP2.GK.AC2 (1.7 ± 1.7 °C) (0 to 7.1 °C) than in COMP2.GK.AC1 (2.6 ± 2.5 ° C) (0.3 to 14.7 °C) (*p* = 0.024). Conversely, on the dorsum of the hands, there was no significant difference (*p* > 0.05). It should be noted that only the COMP2.GK.AC2 workers wore a thermal glove on the dominant hand (66.7%) and, overall, no COMP2.GK.AC workers wore this type of glove on their non-dominant hand.

The group who wore the same number and type of gloves on both hands (GWK1) and the overall group (GWK3) had lower finger temperatures on the dominant hand than the non-dominant hand (*p* < 0.001) (Table 7). The exception was GWK2, which included workers wearing an AC glove on the non-dominant hand and a thermic glove on the dominant hand (*p* > 0.05). This is corroborated by the fact that most workers presented lower average finger temperatures on the dominant hand than the non-dominant hand on both the palm (62.6%) and the dorsum (63.6%).

A total of 46% of workers had at least one finger with an average temperature of ≤15 °C on the dominant hand, while 43.9% and 47.6% of workers presented temperature deltas between the fingers of >1 °C on the palmar and dorsal surfaces, respectively. Additionally, there was no difference in the average deltas of the finger temperatures in the GWK (*p* > 0.05), with a minimum delta value of −14.9 °C and a maximum of 8.2 °C.

## 4. Discussion

The results of the present study were similar to those found by Tirloni et al. [3], in which poultry slaughterhouse workers wore up to five gloves, and the most prevalent amount was three overlapping gloves (57.3%). Another study analyzed each hand individually, which also verified that most of workers wore three gloves on their non-dominant (56.6%) and dominant (60.5%) hands [24].

A review identified that many studies show that gloves may have negative effects on manual dexterity, tactile sensitivity, handgrip strength, muscle activity and fatigue, and comfort [25]. The workers’ choice of three gloves may have occurred, as the NR-36 manual itself states, due to certain tasks in slaughterhouses requiring gloves to protect against humidity, cold, and cuts [9]. However, this regulation does not refer to the number and sequencing of gloves that may be adopted. It is worth mentioning that the use of each glove type may aim to avoid excessively increasing hand size while meeting the hygiene, health, and safety norms required in Brazilian slaughterhouses. Willms et al. [26] reported that perceived exertion significantly increased as a function of glove thickness. Notwithstanding, for GK, most workers did not wear a thermic glove to protect against cold in the hand that handled products (50.3%), and on the other hand, workers wore more than one overlapping waterproof glove (70.6%). 

Nitrile gloves are ideal for handling food and working in refrigerated environments [27], preventing professionals from contaminating the food [28], as they are waterproof. These gloves may be disposable or not, offer chemical [27,29,30] and mechanical resistance [27], and they may be thin or thick. Thin models provide a higher dexterity and offer safety for work that requires tactile sensitivity [27,30]. Perhaps this is the reason that the workers chose to wear two waterproof gloves, with one thermal glove below the waterproof one.

Despite the use of several overlapping gloves, other studies corroborate the findings of the current study (98.1%), as the finger temperatures of workers at poultry slaughterhouses were ≤24 °C [3,4]. According to Lehmuskallio et al. [11], the skin sensation of discomfort due to exposure to cold occurs when the skin temperature is <25 °C. In the encyclopedia of the International Labor Organization, Vogt [10] cites that hand and finger temperatures between 20 and 27 °C decrease high-precision work performance and muscular endurance. On the other hand, it is stated that when temperatures are <15 °C, gross muscle strength is reduced, in addition to deteriorating muscle coordination and causing frequent pain. That is, finger temperatures of ≤ 15 °C should be avoided, since they can cause high physiological strain [22]. Nevertheless, it was found that this condition (≤15 °C) is common in slaughterhouse workers, as was evidenced both in the present study (46.6%) and in previous studies (66.4% and 76%, respectively) [3,4].

In addition to proof by direct measurement that workers’ finger temperatures are cold (thermography), investigations have also shown that most workers feel cold in their hands (78% and 62.5%, respectively) [4,31], contrary to the present study (46.1%). Before the existence of NR-36, worker analyses demonstrated that 59.2% of workers felt cold and confirmed an association between body discomfort and the perception of cold (OR = 2.05; 95% CI 1.44 to 2.91) [32].

In this study, almost half of the workers felt discomfort in their upper limbs, similar to the results of Pinetti and Buczek [33] (43%). However, other studies have indicated that most workers feel bodily discomfort in general (71.5%) [32], as well as in their upper limbs (54%) [24]. Some studies have revealed the shoulder region as the most affected [31,33], as in the present study.

Unrelated to the gloves used by workers, previous studies have proven that there is a difference in the average finger temperatures between the non-dominant and dominant hands for those who use tools [3]. Some authors stated that this may occur due to handling refrigerated (≤7 °C) products (meat) with the non-dominant hand and using a knife in the dominant hand, as well as the use of a CM glove on the non-dominant hand [3]. For Caple [34], the use of a chainmail glove on the non-dominant hand by meat processing industry workers was perceived as uncomfortable and resulted in the cooling of the hand when used in a cold environment.

Research has established that no significant differences are found in the finger temperatures for groups that do not use tools (*p* > 0.05) [3]. These outcomes can be justified by the direct contact and/or handling of cold products with both hands, unlike the present study, in which the workers did not wear thermal gloves on the dominant hand and these fingers were colder than the opposite hand. One option to ensure that both hands have similar temperatures is to provide training for workers to use both hands when performing work tasks.

Conforming to the results of the paper by Tirloni et al. [3], the chance of feeling cold for a worker who used a tool was greater than that for a worker who did not (OR = 3.19, 95% CI 1.46; 6.94), revealing that all workers who used a tool wore a chainmail glove on the non-dominant hand. The current results show that the worker group who wore a CM glove (non-dominant hand) had average finger temperatures lower than the group that wore an AC glove. 

CM gloves were developed to protect users’ hands when working against knife blows and/or similar movements, which are indicated for work with a high risk of cutting and general work involving highly aggressive horizontal and vertical cuts [35]. Overall, CM gloves are recommended for activities that use typical manual knives for meat processing and those similar in slaughterhouses, such as cuts and incisions in deboning, trimming, and the evisceration of cattle, goats, sheep, pigs, poultry, and fish [36]. According to one manufacturer, AC gloves protect against abrasive, scoring, cutting, and piercing agents, as well as cold thermal agents [37]. In addition, there are AC gloves with cut resistance comparable to chainmail gloves, which provide comfort and flexibility with a prolonged durability [38]. AC gloves are recommended for hand protection against medium- and high-intensity mechanical risks when handling and cutting food [39]. Another issue is that one CM glove can cost approximately 6 to 20 times more than a pair of AC gloves (depending on the model) [40]. However, CM gloves have an excellent cost–benefit ratio due to their high resistance [36], as they are made of stainless-steel mesh, with wires with diameters of 0.55 mm [35,36]. The CM glove may interfere with the user’s comfort, as one paper proved that meat workers disliked CM gloves due to their poor comfort and fit for the range of male/female hand sizes, poorly fitting the fingertips and resulting in a lack of grip sensitivity. Additionally, the mesh becomes cold and the weight of the glove fatigues the hand/arm [34]. When the CM glove is larger than the size of the hand, the glove adjuster can be used, which is another device added to the worker’s hand, interfering with its natural movement. 

The comparison between the two groups that used CM gloves on the non-dominant hand showed that the effect of using a thermal glove under the CM glove was positive, considering that the finger temperatures of workers in this group (COMP2.GK.CM2) were higher than those who wore only waterproof gloves under the CM glove (COMP2.GK.CM1).

The NR-36 manual states that the layer closest to the body must be insulating and keep humidity away from the skin to keep it dry. In addition, multiple layers allow for the adjustment of thermal insulation according to the heat production, enabling the maximum trapping of immobile air, which is an excellent thermal insulator [9]. This insulating effect did not efficiently increase finger temperatures, because in the comparison between the GK.AC groups (non-dominant hand), workers who used an extra waterproof glove (insulator) in contact with the skin (COMP2.GK.AC1) did not have finger temperatures significantly higher than those in GK.AC2 (*p* > 0.05).

There was also inappropriate use of the AC gloves, as they were used on both hands by 37.6% of workers who did not use a knife, that is, they were used to protect against the cold only.

AC gloves are knitted in synthetic and stainless-steel yarns coated with polyethylene [37], providing protection against cuts; against cuts and cold [37,38]; and against cuts, cold, and heat, simultaneously [38]. There is one model with polyamide yarns in combination with glass fibers, cotton, polyester, and plastic fibers, in addition to cut resistance at level C (A to F, with A being the worst result) [38].

Both thermic and anti-cut gloves (with thermal protection) protect against cold by convection and contact, but when wet, they can lose their insulating properties against cold [41,42], a factor that must be verified and monitored by workers’ health managers.

Anti-cut gloves with thermal function can be used under waterproof gloves and in low-temperature environments [43]. Using AC gloves without thermal protection is inadequate, because if the worker added another thermal glove under the AC glove (totaling three gloves), the size of the hand would increase significantly, and could consequently interfere with their manual dexterity [7]. One author cited that wearing two gloves was found to be bulky, particularly whilst wearing the thicker AC glove [34]. Therefore, studies must be carried out to verify workers’ perceptions of comfort regarding the contact of the steel wires of anti-cut gloves with the skin, with and without thermal protection.

The anti-cut glove types frequently used in slaughterhouses were present in this study. The workers were asked whether there was a mandatory need for CM gloves, since most tasks (64.7%) involved in the present study did not require the worker to perform knife strikes (trimming, conveyor cuts, and weight adjustment cuts—scales). Additionally, only a few GK workers wore AC gloves on both hands (11.2%). According to Caple [34], reductions of up to 80% of laceration injuries can be anticipated if AC gloves are worn on both hands. Furthermore, he mentioned that knife lacerations account for around 25% of all workers’ compensation costs to the industry, with sprains and strains accounting for over 50% of costs, in which these lacerations included cuts to the non-knife hand and forearm [34]. Despite this, the author conducted a trial that suggested wearing thicker AC gloves or the CM glove on the non-knife hand and the thinner AC glove on the knife hand. This is to prevent the incidence of “run through” lacerations. According to the statistical yearbook of work accidents [44], wrist and hand injury (S-61) was the highest incident cited by the International Classification of Diseases (ICD-11), with fingers being the body region most affected (responsible for 129,299 occupational accidents in 2022 out of a total of 648,366).

It should be noted that companies are required to provide employees with PPE appropriate to the risk and in perfect functioning condition, and workers are responsible for using it [6]. It was observed in a technical report that the validity of the CM glove is indeterminate [36], however, for another supplier, the validity of the gloves was described on the product packaging [38]. On the other hand, the durability of the gloves depends on the employer and the worker, as manufacturers describe guidelines to be followed regarding the proper use and conservation of PPE. Furthermore, in accordance with NR-6, the company must guide and train the worker on the appropriate use, storage, and conservation of PPE, along with replacing it immediately when damaged or lost [6].

Suggestions regarding the sequencing of gloves for those who use a knife or not have been made, however, the biological and physical individuality of each worker must be considered, in addition to other variables. According to Lehmuskallio et al. [11], skin sensations depend on several factors other than skin temperature, such as the environmental temperature preceding the cold exposure, skin region, individual cold sensitivity, cold adaptation, etc. Besides this, the NR-36 recommendations [5] must be followed, in which gloves must be compatible with the nature of the tasks, the environmental conditions, and workers’ hand sizes, in addition to being replaced when necessary, in order to avoid compromising their effectiveness. As for tool use, the standard mentions that the type, shape, and texture of the knife handles must be appropriate to the task, workers’ hands, and possible glove use.

Therefore, it is proposed that the health and safety teams of slaughterhouses carry out detailed analyses of tasks and job rotations (if cutting tasks are involved). This action is intended to determine the types, quantity, and sequencing of gloves required on both hands, not only prioritizing the cost x benefit in the acquisition of PPE, but mainly considering the protection of workers against cuts, cold, and humidity, as well as comfort in use. Ultimately, OSHA [2] cites that when employees work in a cold environment, employers should limit their exposure to cold by providing a warm, dry break area and allowing frequent, short breaks to let workers warm up.

Finally, based on the results of the present study and following the recommendations regarding the three types of protection (when necessary), in an artificially cold environment, the glove sequences that can promote greater thermal protection for those who use or do not use a knife are described in Table 8.

### Strengths and Limitations

Some limitations were noted as follows: in two glove sequence groups, the sample of workers was reduced, the same workers with different types of gloves were not tested, the same workers with different glove sequences were not tested, each company adopted gloves of different brands/models, and these were categorized in this study according to their general characteristics. Additionally, no information was collected on the guidelines provided by the health and safety teams of the slaughterhouses and the reasons why they preferred the sequencing they were wearing at the time of collection. However, as a strength, this was an exploratory study, and is the first one to analyze the effects of the sequencing, number, and types of gloves on the finger temperatures of poultry slaughterhouse workers.

## 5. Conclusions

This study showed that workers wore from one to five overlapping gloves, but the use of three gloves was most common. Even when wearing several gloves, most workers had fingers with temperatures of ≤24 °C. Additionally, almost half of the workers felt cold in at least one hand, had at least one finger with an average temperature of ≤15 °C, and felt bodily discomfort in their upper limbs, mainly in the shoulder and wrist.

Most poultry slaughterhouses provide chainmail gloves for their workers to protect against cuts, but it has been shown that this glove on the non-dominant hand significantly lowers workers’ finger temperatures compared to when wearing the anti-cut glove. Furthermore, the chance of a worker who wore a chainmail glove to have average finger temperatures of ≤15 °C was greater than that for a worker who wore an anti-cut glove. These results confirm the effect of wearing a thermal glove under the chainmail glove, as finger temperatures were significantly greater.

In the group that used a knife, it was found that the non-dominant hand (product) was colder in most groups and hand surfaces in relation to the dominant hand (knife). Conversely, workers who did not use a knife also demonstrated their product-handling hand to have lower temperatures, but this was the dominant hand.

The findings of the present study indicate that the quantity, type, and sequencing of gloves determine the finger temperatures and thermal comfort of poultry slaughterhouse workers. Therefore, the health and safety teams of these companies must consider these issues when determining the appropriate PPE for each work activity in cold environments.

## Figures and Tables

**Figure 1 ijerph-21-01314-f001:**
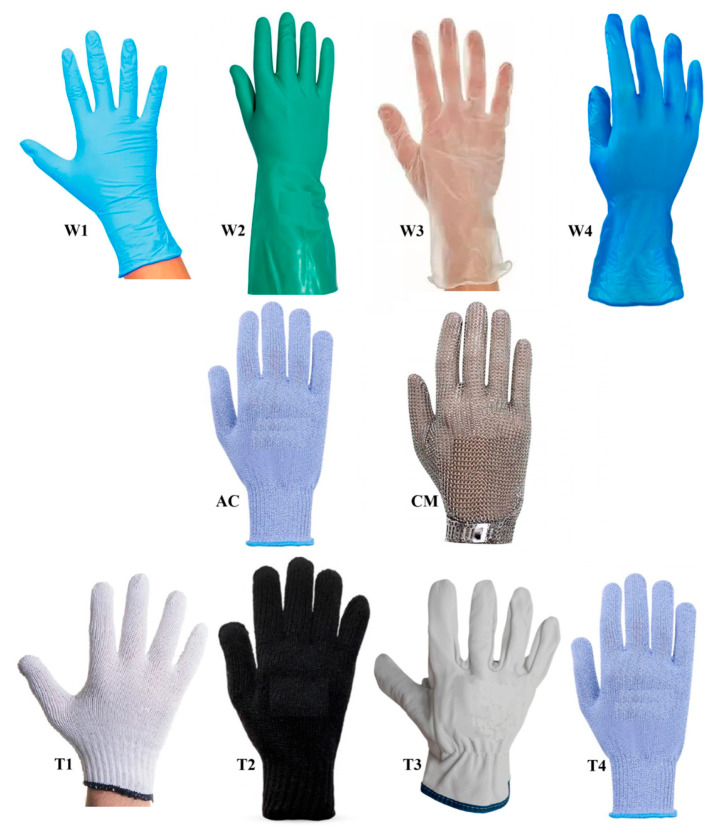
Types of overlapping gloves worn [waterproof protection: nitrile (W1), thick nitrile (W2), plastic (W3), and thick plastic (W4); cut protection: anti-cut (AC) and chainmail (CM); and thermic/cold protection: cotton (T1), wool (T2), leather (T3), and anti-cut (T4)].

**Figure 2 ijerph-21-01314-f002:**
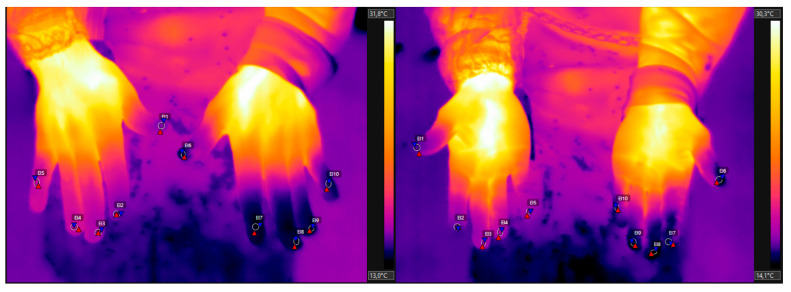
Thermographic images of the hand palm, dorsum, and finger areas of a worker that used a knife.

**Figure 3 ijerph-21-01314-f003:**
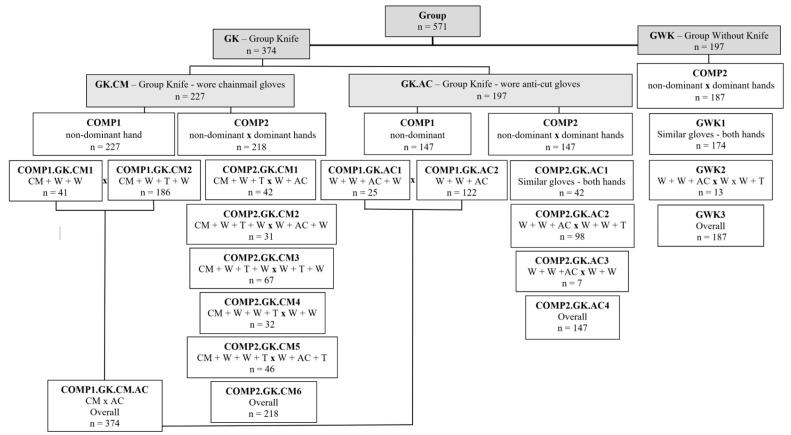
Description of the study groups according to knife use and types and sequences of gloves worn (GK—“Group knife”; GWK—“Group without knife”; Glove types: CM—Chainmail, W—Waterproof, AC—Anti-cut, and T—Thermic; COMP1—Comparison between the glove sequences on non-dominant hands; COMP2—Comparison between the glove sequences on non-dominant hands X dominant hand; *n* = number of workers in relation to the sequencing of gloves worn; Similar gloves—means that the types of gloves used on the left hand were the same as the types of gloves on the right hand, with different overlapping gloves worn).

**Table 1 ijerph-21-01314-t001:** Organizational work characteristics of slaughterhouses.

Slaughterhouses	Workers (*n*)	Chickens Slaughtered Daily	Rest Breaks/Daily	Duration of Work Shift	Participants (*n*)
1	1200	100,000	3 × 20 min	8 h 48 min	52
2	3500	240,000	4 × 10 min	7 h 20 min	136
3	1500	150,000	4 × 15 min	8 h 48 min	96
4	1000	112,000	2 × 20 min	8 h 48 min	119
5	2000	180,000	5 × 10 min	7 h 20 min	84
6	326	111,000	3 × 20 min	8 h 48 min	44
7	1130	115,000	3 × 20 min	8 h 48 min	40
**Total**					571

**Table 2 ijerph-21-01314-t002:** Relation between knife use and organizational work characteristics, cold perception, and musculoskeletal discomfort of slaughterhouse workers.

Variables	With Knife (*n* = 374)	Without Knife (*n* = 197)	Overall (*n* = 571)
**Work Shifts**	**(*n*, %)**	**(*n*, %)**	**(*n*, %)**
Morning	166 (44.4)	134 (68.0)	300 (52.5)
Afternoon	208 (55.6)	63 (32.0)	271 (47.5)
**Gender**			
Female	313 (83.7)	89 (45.2)	402 (70.4)
Male	61 (16.3)	108 (54.8)	169 (29.6)
**Cold perception of at least one hand**			
No	179 (47.9)	129 (65.5)	308 (53.9)
Yes	195 (52.1)	68 (34.5)	263 (46.1)
**Finger temperature #**			
**≤15 °C**			
No	202 (54.0)	103 (52.3)	305 (53.4)
Yes	172 (46.0)	94 (47.7)	266 (46.6)
**≤24 °C**			
No	5 (1.3)	6 (3.0)	11 (1.9)
Yes	369 (98.7)	191 (97.0)	560 (98.1)
**Musculoskeletal discomfort**			
No	171 (45.7)	115 (58.4)	286 (50.1)
Yes	203 (54.3)	82 (41.6)	285 (49.9)
**Bodily region**			
**Shoulder**			
No	246 (65.8)	137 (69.5)	383 (67.1)
Yes	128 (34.2)	60 (30.5)	188 (32.9)
**Arm**			
No	355 (94.9)	194 (98.5)	549 (96.1)
Yes	19 (5.1)	3 (1.5)	20 (3.9)
**Elbow**			
No	360 (96.3)	191 (97.0)	551 (96.5)
Yes	14 (3.7)	6 (3.0)	20 (3.5)
**Forearm**			
No	361 (96.5)	192 (97.5)	553 (96.8)
Yes	13 (3.5)	5 (2.5)	18 (3.2)
**Wrist**			
No	320 (85.6)	179 (90.9)	499 (87.4)
Yes	54 (14.4)	18 (9.1)	72 (12.6)
**Hand**			
No	327 (87.4)	185 (93.9)	512 (89.7)
Yes	47 (12.6)	12 (6.1)	59 (10.3)
**Fingers**			
No	350 (93.6)	191 (97.0)	541 (94.7)
Yes	24 (6.4)	6 (3.0)	30 (5.3)

# At least one finger ≤ the temperature parameter, regardless of hand surfaces.

**Table 3 ijerph-21-01314-t003:** Description of the gloves and the perception of cold in the hands on both body sides and with knife use.

	GK (*n* = 374)		GWK (*n* = 197)		Overall (*n* = 571)	
Number of Gloves	Non-Dominant (*n*, %)	Dominant (*n*, %)	Non-Dominant (*n*, %)	Dominant (*n*, %)	Non-Dominant (*n*, %)	Dominant (*n*, %)
No	0 (0)	0 (0)	15 (7.6)	10 (5.1)	15 (2.6)	10 (1.8)
Yes	374 (100.0)	374 (100.0)	182 (92.4)	187 (94.9)	556 (97.4)	561 (98.2)
1 glove	0 (0)	26 (7.0)	32 (16.2)	35 (17.7)	32 (5.6)	61 (10.7)
2 gloves	21 (5.6)	115 (30.7)	61 (31.0)	66 (33.5)	82 (14.4)	181 (31.7)
3 gloves	232 (62.0)	210 (56.1)	71 (36.1)	66 (33.5)	303 (53.1)	276 (48.3)
4 gloves	119 (31.8)	23 (6.1)	18 (9.1)	20 (10.2)	137 (24.0)	43 (7.5)
5 gloves	2 (0.5)	0 (0)	0 (0)	0 (0)	2 (0.4)	0 (0)
**Types of Gloves**						
Without gloves	0 (0.0)	0 (0.0)	15 (7.6)	10 (5.1)	15 (2.6)	10 (1.8)
Chainmail	227 (60.7)	0 (0.0)	2 (1.0)	0 (0.0)	229 (40.1)	0 (0)
Anti-cut	147 (39.3)	126 (33.7)	69 (35.0)	67 (34.0)	216 (37.8)	193 (33.8)
Waterproof	374 (100.0)	374 (100.0)	178 (90.4)	182 (92.4)	552 (96.7)	556 (97.4)
Thermic	190 (50.8)	170 (43.9)	86 (43.7)	74 (37.6)	276 (48.3)	244 (42.7)
**Function of protective gloves**						
Without protection	0 (0.0)	0 (0.0)	15 (7.6)	10 (5.1)	15 (2.6)	10 (1.8)
Cuts, humidity, and cold	337 (90.1)	126 (33.7)	70 (35.5)	66 (33.5)	407 (71.3)	192 (33.6)
Cuts and humidity	37 (9.9)	0 (0.0)	0 (0.0)	0 (0.0)	37 (6.5)	0 (0.0)
Cuts and cold	0 (0.0)	0 (0.0)	1 (0.5)	1 (0.5)	1 (0.2)	1 (0.2)
Humidity and cold	0 (0.0)	164 (43.9)	81 (41.1)	70 (35.5)	81 (14.2)	234 (41.0)
Humidity	0 (0.0)	84 (22.5)	27 (13.7)	47 (23.9)	27 (4.7)	131 (22.9)
Cold	0 (0.0)	0 (0.0)	3 (1.5)	3 (1.5)	3 (0.5)	3 (0.5)
**Perception of cold**						
No	190 (50.8)	239 (63.9)	131 (66.5)	130 (66.0)	321 (56.2)	369 (64.6)
Yes	184 (49.2)	135 (36.1)	66 (33.5)	67 (34.0)	250 (43.8)	202 (35.4)
Mild	52 (13.9)	71 (19.0)	24 (12.2)	25 (12.7)	76 (13.3)	96 (16.8)
Moderate	97 (25.9)	54 (14.4)	30 (15.2)	28 (14.2)	127 (22.2)	82 (14.4)
Severe	35 (9.4)	10 (2.7)	12 (6.1)	14 (7.1)	47 (8.2)	24 (4.2)

GK—“Group knife”; GWK—“Group without knife”; GK dominant hand—the hand in which the worker uses a knife; and Types of gloves and function of protective gloves—regarding the classification and main function of the glove used. Percentage is in relation to the total in each group.

**Table 4 ijerph-21-01314-t004:** Finger temperature difference between diverse groups of chainmail and anti-cut glove sequences on the non-dominant hands.

COMP 1													
	GK.CM1		GK.CM2			GK.AC1		GK.AC2			GK Overall		
Glove Combinations	CM + W + W		CM + W + T + W			W + W + AC + W		W + W + AC			GK.CM	GK.AC	
Palmar Surface	Mean ± SD	Min.–Max. (°C)	Mean ± SD	Min–Max (°C)	*p*	Mean ± SD	Min.–Max. (°C)	Mean ± SD	Min–Max (°C)	*p*	Mean ± SD	Mean ± SD	*p*
Finger 1	16.6 ± 3.1	11.2–24.7	18.6 ± 3.6	11.1–30.6	0.001 *	18.9 ± 3.5	14–27.5	18.8 ± 3.3	12.1–29.1	0.935	18.3 ± 3.6	18.9 ± 3.3	0.108
Finger 2	15.0 ± 2.8	9.8–22.4	17.2 ± 3.1	9.6–30.5	<0.001 *	17.2 ± 2.8	12.7–23.7	17.6 ± 3.0	11.8–28.2	0.544	16.8 ± 3.2	17.6 ± 3.0	0.016 *
Finger 3	14.8 ± 2.7	8.8–20.2	16.9 ± 3.1	8.9–30.2	<0.001 *	17.3 ± 2.9	12.3–24.2	17.5 ± 3.2	11.9–28.9	0.810	16.5 ± 3.2	17.5 ± 3.2	0.005 *
Finger 4	15.2 ± 2.8	9.9–24.1	17.6 ± 3.5	9.3–30.6	<0.001 *	18.2 ± 4.0	12.7–27.2	17.8 ± 3.4	12.7–30.0	0.566	17.1 ± 3.5	17.8 ± 3.5	0.053
Finger 5	15.2 ± 2.7	10.9–22.1	18.1 ± 3.9	10.2–30.8	<0.001 *	18.6 ± 4.4	13.3–28.9	18.2 ± 3.8	12.8–31.7	0.664	17.6 ± 3.8	18.3 ± 3.9	0.086
Overall	15.3 ± 2.7	10.6–22.7	17.7 ± 3.2	9.8–30.5	<0.001 *	18.1 ± 3.3	13.1–24.9	18.0 ± 3.2	13.0–28.9	0.930	17.3 ± 3.9	18.0 ± 3.2	0.029 *
**Dorsal Surface**													
Finger 1	16.6 ± 3.2	11.2–25.1	18.7 ± 3.4	10.9–29.5	<0.001 *	18.8 ± 3.3	14.2–26.5	18.5 ± 3.1	12.9–29.4	0.655	18.3 ± 3.4	18.6 ± 3.2	0.456
Finger 2	14.8 ± 2.7	9.6–22.2	16.7 ± 3.1	10.1–28.0	<0.001 *	16.9 ± 2.5	13.1–22.6	17.0 ± 2.9	11.7–26.2	0.896	16.3 ± 3.1	16.9 ± 2.8	0.054
Finger 3	14.4 ± 2.7	9.2–21.2	16.3 ± 3.0	9.6–28.3	<0.001 *	17.0 ± 2.7	14.2–26.5	16.8 ± 3.1	11.3–26.8	0.755	16.0 ± 3.0	16.8 ± 3.0	0.011 *
Finger 4	14.6 ± 2.6	9.7–22.6	16.9 ± 3.2	9.8–29.2	<0.001 *	17.7 ± 3.7	13.1–22.6	17.1 ± 3.3	12.3–27.7	0.417	16.4 ± 3.2	17.2 ± 3.3	0.033
Finger 5	14.9 ± 2.6	10.6–22.1	17.6 ± 3.6	10.8–28.5	<0.001 *	18.1 ± 4.1	12.7–27.5	17.6 ± 3.6	12.7–30.2	0.564	17.1 ± 3.6	17.7 ± 3.7	0.118
Overall	15.1 ± 2.6	10.2–22.1	17.2 ± 3.1	10.4–28.3	<0.001 *	17.7 ± 3.1	12.9–23.8	17.4 ± 3.1	12.6–27.8	0.656	16.8 ± 3.1	17.4 ± 3.1	0.064
**Number of gloves worn—n (%)**													
2	17 (41.5)		0 (0)			0 (0)		4 (3.3)			17 (7.5)	4 (2.7)	
3	20 (48.8)		89 (47.8)			11 (44.0)		112 (91.8)			109 (48.0)	123 (83.7)	
4	4 (9.8)		95 (51.1)			14 (56.0)		6 (4.9)			99 (43.6)	20 (13.6)	
5	0 (0)		2 (1.1)			0 (0)		0 (0)			2 (0.9)	0 (0)	
**Non-dominant hand**													
≤ 15 °C	29 (70.7)		87 (46.8)			8 (32.0)		40 (32.8)			116 (51.1)	48 (32.7)	
≤ 24 °C	41 (100)		183 (98.4)			25 (100)		119 (97.5)			224 (98.7)	144 (98.0)	
**Total**	**41 (18.1)**		**186 (81.9)**			**25 (17.0)**		**122 (83.0)**			**227 (100)**	**147 (100)**	

Independent Student *t*-test; * *p* ≤ 0.05; the non-dominant hand manipulated the products; Glove types: CM—Chainmail, W—Waterproof, AC—Anti-cut, and T—Thermic; GKCM—wore chainmail gloves = 227 workers; GK.AC—wore anti-cut gloves = 147 workers; GK = 374 workers.

**Table 5 ijerph-21-01314-t005:** Comparison between the mean temperatures of the fingers of the non-dominant and dominant hands for different sequencing of the gloves worn by the knife group.

	Comp2.GK.CM1			Comp2.GK.CM2			Comp2.GK.CM3			Comp2.GK.CM4			Comp2.GK.CM5			Comp2.GK.CM6		
Glove Combinations	CM + W + T	W + AC		CM + W + T + W	W + AC +W		CM + W + T +W	W + T + W		CM + W + W	W + W		CM + W + W + T	W + AC + T		Overall		
	Non- Dominant	Dominant (knife)		Non- Dominant	Dominant (Knife)		Non- Dominant	Dominant (Knife)		Non- Dominant	Dominant (Knife)		Non- Dominant	Dominant (Knife)		Non- Dominant	Dominant (Knife)	
	Mean ± SD		*p*	Mean ± SD		*p*	Mean ± SD		*p*	Mean ± SD		*p*	Mean ± SD		*p*	Mean ± SD		*p*
Finger 1	18.6 ± 3.1	19.4 ± 3.3	0.094	19.5 ± 4.2	19.6 ± 4.0	0.828	18.8 ± 3.5	19.8 ± 4.2	0.003 *	16.2 ± 3.2	17.4 ± 3.5	0.009 *	17.9 ± 3.7	19.0 ± 3.6	0.003 *	18.3 ± 3.6	19.2 ± 3.8	<0.001 *
Finger 2	17.2 ± 2.1	18.3 ± 3.1	0.006 *	17.8 ± 4.2	18.9 ± 4.2	0.004 *	17.2 ± 3.0	19.2 ± 4.5	<0.001 *	14.6 ± 2.9	17.0 ± 3.6	<0.001 *	16.6 ± 3.2	18.0 ± 3.5	<0.001 *	16.8 ± 3.2	18.4 ± 3.9	<0.001 *
Finger 3	17.0 ±2.5	18.3 ± 3.4	0.011 *	17.2 ± 3.7	18.4 ± 3.8	<0.001 *	17.1 ± 3.3	18.8 ± 4.5	<0.001 *	14.4 ± 2.9	16.9 ± 3.6	<0.001 *	16.4 ± 3.1	18.4 ± 3.9	<0.001 *	16.5 ± 3.2	18.3 ± 4.0	<0.001 *
Finger 4	17.4 ± 2.9	19.1 ± 3.8	0.002 *	18.5 ± 4.2	19.4 ± 4.2	0.083	17.6 ± 3.5	19.7 ± 4.9	<0.001 *	14.8 ± 3.0	17.5 ± 3.6	<0.001 *	17.0 ± 3.3	19.1 ± 3.7	<0.001 *	17.1 ± 3.5	19.1 ± 4.2	<0.001 *
Finger 5	17.9 ± 3.0	19.3 ± 3.6	<0.001 *	19.2 ± 4.6	19.9 ± 4.1	0.184	18.1 ± 3.9	20.1 ± 4.9	<0.001 *	14.8 ± 2.8	17.9 ± 3.4	<0.001 *	17.6 ± 3.9	19.2 ± 3.8	<0.001 *	17.6 ± 3.9	19.4 ± 4.2	<0.001 *
All fingers	17.6 ± 2.4	18.9 ± 3.1	0.001 *	18.4 ± 4.0	19.2 ± 3.8	0.021 *	17.8 ± 3.2	19.5 ± 4.4	<0.001 *	14.9 ± 2.8	17.3 ± 3.4	<0.001 *	17.1 ± 3.3	18.7 ± 3.5	<0.001 *	17.3 ± 3.3	18.9 ± 3.8	<0.001 *
**Dorsal Surface**																		
Finger 1	18.3 ± 2.6	19.7 ± 3.3	0.005 *	19.5 ± 4.1	19.6 ± 3.8	0.881	18.8 ± 3.3	20.3 ± 4.5	<0.001 *	16.2 ± 3.3	17.5 ± 3.6	0.006	18.3 ± 3.6	19.0 ± 3.6	0.028 *	18.3 ± 3.5	19.4 ± 3.9	<0.001 *
Finger 2	16.4 ± 2.1	18.0 ± 3.1	<0.001 *	17.6 ± 4.1	18.5 ± 4.0	0.019 *	16.7 ± 2.9	18.8 ± 4.4	<0.001 *	14.5 ± 2.9	17.0 ± 4.0	<0.001 *	16.3 ± 3.2	17.9 ± 3.8	<0.001 *	16.4 ± 3.1	18.2 ± 3.9	<0.001 *
Finger 3	16.2 ± 2.4	17.7 ± 3.2	0.003 *	16.8 ± 3.5	18.0 ± 3.6	0.004 *	16.5 ± 3.0	18.3 ± 4.4	<0.001 *	14.1 ± 2.9	16.9 ± 4.0	<0.001 *	15.8 ± 2.9	17.8 ± 3.8	<0.001 *	16.0 ± 3.0	17.8 ± 3.9	<0.001 *
Finger 4	16.5 ± 2.3	18.4 ± 3.7	<0.001 *	17.9 ± 4.1	18.9 ± 4.1	0.036 *	16.9 ± 3.3	19.1 ± 4.6	<0.001 *	14.4 ± 2.8	17.2 ± 3.8	<0.001 *	16.5 ± 3.1	18.3 ± 3.8	<0.001 *	16.5 ± 3.3	18.5 ± 4.1	<0.001 *
Finger 5	17.3 ± 2.9	18.9 ± 3.6	<0.001 *	18.6 ± 4.3	19.7 ± 4.1	0.025 *	17.7 ± 3.7	19.7 ± 4.9	<0.001 *	14.7 ± 2.7	17.5 ± 3.3	<0.001 *	17.1 ± 3.6	18.8 ± 3.8	<0.001 *	17.2 ± 3.6	19.0 ± 4.1	<0.001 *
All fingers	16.9 ± 2.3	18.5 ± 3.2	<0.001 *	18.1 ± 3.8	18.9 ± 3.7	0.013 *	17.3 ± 3.0	19.3 ± 4.4	<0.001 *	14.8 ± 2.8	17.2 ± 3.6	<0.001 *	16.8 ± 3.2	18.3 ± 3.7	<0.001 *	16.9 ± 3.1	18.6 ± 3.8	<0.001 *
**Number of gloves worn—n (%)**																		
1	0 (0)	0 (0)		0 (0)	0 (0)		0 (0)	0 (0)		0 (0)	15 (46.9)		0 (0)	11 (23.9)		0 (0)	0 (0)	
2	0 (0)	25 (59.5)		0 (0)	0 (0)		0 (0)	25 (37.3)		12 (37.5)	15 (46.9)		0 (0)	34 (73.9)		12 (5.5)	26 (11.9)	
3	36 (85.7)	15 (35.7)		1 (3.2)	31 (100)		31 (46.3)	41 (61.2)		16 (50.0)	2 (6.2)		21 (45.7)	1 (2.2)		105 (48.2)	99 (54.4)	
4	5 (11.9)	2 (4.8)		30 (96.8)	0 (0)		35 (52.2)	1 (1.5)		4 (12.5)	0 (0)		25 (54.3)	0 (0)		99 (45.4)	90 (41.3)	
5	1 (2.4)	0 (0)		0 (0)	0 (0)		1 (1.5)	0 (0)		0 (0)	0 (0)		0 (0)	0 (0)		2 (0.9)	3 (1.4)	
**On each hand**																		
≤15 °C	18 (42.9)	8 (19.0)		13 (41.9)	8 (25.8)		29 (43.3)	17 (25.4)		24 (75.0)	18 (56.3)		27 (58.7)	16 (34.8)		111 (50.9)	67 (30.7)	
≤24 °C	42 (100)	42 (100)		30 (96.8)	28 (90.3)		65 (97.0)	61 (91.0)		32 (100)	31 (96.9)		46 (100)	44 (95.7)		215 (98.6)	206 (94.5)	
**Overall †**																		
≤15 °C	19 (45.2)			14 (45.2)			31 (46.3)			24 (75.9)			28 (60.9)			116 (53.2)		
≤24 °C	42 (100)			30 (96.8)			65 (97.0)			32 (100)			46 (100)			215 (98.6)		
**Total *n* (%) #**	**42 (19.3)**			**31 (14.2)**			**67 (30.7)**			**32 (14.7)**			**46 (21.1)**			**218 (100)**		

# Number of workers and % related to row; † Overall is at least one finger of hands; Glove types: CM—Chainmail, W—Waterproof, AC—Anti-cut, and T—Thermic; Paired Student’s *t*-test; * *p* ≤ 0.05.

**Table 6 ijerph-21-01314-t006:** Comparison between the workers’ finger temperatures when using anti-cut gloves on the non-dominant hand versus the glove sequence on the dominant hand.

	Comp2.GK.AC1			Comp2.GK.AC2			Comp2.GK.AC3			Comp2.GK.AC4		
Glove Combinations	W + W + AC + W or W + W + AC			W + W + AC	W + W + T		W + W + AC	W + W		Overall		
	Non- Dominant	Dominant (Knife)		Non- Dominant	Dominant (Knife)		Non- Dominant	Dominant (Knife)		Non- Dominant	Dominant (Knife)	
Palmar Surface	Mean ± SD		*p*	Mean ± SD		*p*	Mean ± SD		*p*	Mean ± SD		*p*
Finger 1	19.4 ± 3.5	21.8 ± 4.9	<0.001 *	18.5 ± 3.2	19.4 ± 3.9	<0.001 *	19.8 ± 3.2	20.7 ± 5.1	0.358	18.9 ± 3.3	20.2 ± 4.4	<0.001 *
Finger 2	17.9 ± 3.3	20.3 ± 5.0	<0.001 *	17.5 ± 2.9	18.8 ± 3.9	<0.001 *	17.2 ± 2.3	19.2 ± 3.9	0.062	17.6 ± 3.0	19.2 ± 4.3	<0.001 *
Finger 3	18.2 ± 3.5	20.6 ± 4.9	<0.001 *	17.2 ± 3.0	18.4 ± 3.8	<0.001 *	16.9 ± 2.5	19.4 ± 3.3	0.067	17.5 ± 3.2	19.1 ± 4.2	<0.001 *
Finger 4	18.7 ± 4.0	21.3 ± 5.1	<0.001 *	17.6 ± 3.3	18.6 ± 3.8	<0.001 *	16.7 ± 2.6	21.4 ± 4.5	0.009 *	17.8 ± 3.5	19.5 ± 4.4	<0.001 *
Finger 5	19.2 ± 4.4	21.2 ± 5.2	<0.001 *	18.0 ± 3.8	19.4 ± 4.2	<0.001 *	17.1 ± 1.7	21.0 ± 4.4	0.023 *	18.3 ± 3.9	19.3 ± 4.5	<0.001 *
All fingers	18.7 ± 3.6	21.0 ± 4.9	<0.001 *	17.8 ± 3.1	18.9 ± 3.7	<0.001 *	17.5 ± 2.2	20.3 ± 4.2	0.029 *	18.0 ± 3.2	18.9 ± 4.2	<0.001 *
**Dorsal Surface**												
Finger 1	19.4 ± 3.4	22.2 ± 4.7	<0.001 *	18.2 ± 3.1	19.6 ± 3.6	<0.001 *	18.5 ± 2.0	19.0 ± 3.9	0.687	18.6 ± 3.2	20.3 ± 4.1	<0.001 *
Finger 2	17.5 ± 3.1	20.3 ± 4.9	<0.001 *	16.7 ± 2.7	18.4 ± 3.8	<0.001 *	16.7 ± 1.9	18.0 ± 3.6	0.213	16.9 ± 2.8	18.9 ± 4.2	<0.001 *
Finger 3	17.8 ± 3.4	20.5 ± 4.9	<0.001 *	16.5 ± 2.8	17.8 ± 3.7	<0.001 *	15.9 ± 2.0	17.8 ± 3.3	0.219	16.8 ± 3.0	18.6 ± 4.2	<0.001 *
Finger 4	18.3 ± 4.0	20.9 ± 5.2	<0.001 *	16.8 ± 3.0	17.8 ± 3.6	0.002 *	16.0 ± 2.1	18.1 ± 4.2	0.208	17.2 ± 3.3	18.7 ± 4.3	<0.001 *
Finger 5	18.7 ± 4.1	21.2 ± 5.3	<0.001 *	17.4 ± 3.5	18.8 ± 3.9	<0.001 *	16.4 ± 1.7	19.0 ± 4.4	0.146	17.7 ± 3.7	19.5 ± 4.5	<0.001 *
All fingers	18.3 ± 3.4	21.0 ± 4.9	<0.001 *	17.1 ± 2.9	18.5 ± 3.5	<0.001 *	16.7 ± 1.8	18.4 ± 3.7	0.209	17.4 ± 3.1	19.2 ± 4.1	<0.001 *
**Number of gloves worn—n (%)**												
2	2 (4.8)	2 (4.8)		2 (2.0)	2 (2.0)		0 (0)	7 (100)		4 (2.7)	11 (7.5)	
3	25 (59.5)	25 (59.5)		91 (92.9)	92 (93.9)		7 (100)	0 (0)		123 (83.7)	117 (79.6)	
4	15 (35.7)	15 (35.7)		5 (5.1)	4 (4.1)		0 (0)	0 (0)		20 (13.6)	19 (12.9)	
**On each hand**												
≤15 °C	10 (23.8)	9 (21.4)		35 (35.7)	21 (21.4)		3 (42.9)	2 (28.6)		48 (32.7)	32 (21.8)	
≤24 °C	41 (97.6)	36 (85.7)		96 (98.0)	94 (95.9)		7 (100.0)	7 (100.0)		144 (98.0)	137 (93.2)	
**Overall †**												
≤15 °C	11 (26.2)			36 (36.7)			4 (57.1)			51 (34.7)		
≤24 °C	41 (97.6)			97 (99.0)			7 (100.0)			145 (98.6)		
**Total *n* (%) #**	**42 (28.6)**			**98 (66.7)**			**7 (4.8)**			**147 (100)**		

# Number of workers, % related to row; On each hand = palmar and dorsal surfaces; † Overall is at least one finger of the hands; Glove types: CM—Chainmail, W—Waterproof, AC—Anti-cut, and T—Thermic; Paired Student’s *t*-test; * *p* ≤ 0.05; Comp2.GK.AC1—Equal gloves on both hands.

**Table 7 ijerph-21-01314-t007:** Comparison of finger temperatures when wearing different glove sequences for workers who did not use a knife.

	GWK1			GWK2			GWK3		
Glove Sequences	Equal Gloves on Both Hands			W + W + AC	W + W + T		Overall		
	Non- Dominant	Dominant		Non- Dominant	Dominant		Non- Dominant	Dominant	
Palmar Surface	Mean ± SD		*p*	Mean ± SD		*p*	Mean ± SD		*p*
Finger 1	19.5 ± 4.6	19.0 ± 4.7	0.003 *	21.4 ± 4.2	20.5 ± 4.0	0.455	19.6 ± 4.6	19.1 ± 4.7	0.003 *
Finger 2	18.0 ± 4.3	17.7 ± 4.3	0.031 *	19.2 ± 3.5	19.1 ± 3.8	0.938	18.1 ± 4.2	17.8 ± 4.3	0.048 *
Finger 3	17.9 ± 4.3	17.4 ± 4.2	0.002 *	19.5 ± 3.6	19.2 ± 3.6	0.782	18.0 ± 4.3	17.5 ± 4.2	0.003 *
Finger 4	18.3 ± 4.6	17.8 ± 4.6	0.003 *	21.0 ± 3.6	19.7 ± 3.9	0.235	18.5 ± 4.6	17.9 ± 4.6	0.001 *
Finger 5	18.7 ± 4.9	18.2 ± 5.0	<0.011 *	21.5 ± 4.1	18.6 ± 4.1	0.072	18.9 ± 4.9	18.2 ± 4.9	0.002 *
All fingers	18.5 ± 4.4	18.0 ± 4.4	0.001 *	20.5 ± 3.6	19.4 ± 3.3	0.295	18.6 ± 4.4	18.1 ± 4.4	<0.001 *
**Dorsal Surface**									
Finger 1	19.5 ± 4.5	19.1 ± 4.5	0.023 *	20.5 ± 3.9	19.9 ± 3.5	0.591	19.6 ± 4.5	19.2 ± 4.4	0.021 *
Finger 2	18.1 ± 4.4	17.7 ± 4.4	0.015 *	19.2 ± 4.0	19.2 ± 3.9	0.939	18.1 ± 4.4	17.8 ± 4.3	0.023 *
Finger 3	18.0 ± 4.5	17.2 ± 4.3	<0.001 *	19.3 ± 3.9	19.4 ± 4.2	0.971	18.0 ± 4.4	17.4 ± 4.3	<0.001 *
Finger 4	18.1 ±4.6	17.5 ± 4.5	<0.001 *	20.0 ± 3.7	19.3 ± 4.0	0.515	18.3 ± 4.6	17.6 ± 4.5	<0.001 *
Finger 5	18.4 ± 4.7	17.9 ± 4.8	0.001 *	20.5 ± 4.0	18.2 ± 4.0	0.151	18.6 ± 4.7	17.9 ± 4.8	<0.001 *
All fingers	18.4 ± 4.4	17.9 ± 4.4	<0.001 *	19.9 ± 3.7	19.2 ± 3.3	0.475	18.5 ± 4.4	18.0 ± 4.3	<0.001 *
**Number of gloves worn—n (%)**									
0	10 (5.7)	10 (5.7)		0 (0)	0 (0)		10 (5.3)	10 (5.3)	
1	30 (17.2)	30 (17.2)		0 (0)	0 (0)		30 (16.2)	30 (16.2)	
2	61 (35.1)	61 (35.1)		0 (0)	0 (0)		61 (32.6)	61 (32.6)	
3	56 (32.2)	55 (31.6)		12 (92.3)	11 (84.6)		68 (36.4)	66 (35.3)	
4	17 (9.8)	18 (10.3)		1 (7.7)	2 (15.4)		18 (9.6)	20 (10.7)	
**On each hand**									
≤15 °C	72 (41.4)	82 (47.1)		1 (7.7)	4 (30.8)		73 (39.0)	86 (46.0)	
≤24 °C	162 (93.1)	164 (93.0)		11 (84.6)	13 (100)		173 (92.5)	177 (94.7)	
**Overall †**									
≤15 °C	86 (49.4)			4 (30.8)			90 (48.1)		
≤24 °C	169 (97.1)			13 (100)			182 (97.3)		
**Total *n* (%) #**	**174 (93.0)**	**174 (93.0)**		**13 (7.0)**	**13 (7.0)**		**187 (100)**	**187 (100)**	

# Number of workers, % related to row; On each hand = palmar and dorsal surfaces; † Overall is at least one finger of the hands.

**Table 8 ijerph-21-01314-t008:** Glove sequences that can promote greater thermal protection for those who use or do not use a knife.

Tool/Objective	Hand	Glove Sequences (from Outer to Inner)			
Knife—against abrasive, scoring, cutting, and piercing agents and against cold thermal agents.	Non- and/or dominant	Waterproof	Anti-cut **		
		Waterproof	Anti-cut **	Waterproof *	
Knife—against knife blows and/or similar movements #	Non- dominant	Chainmail	Waterproof	Thermic	
	Non- dominant	Chainmail	Waterproof	Thermic	Waterproof *
Without knife—against humidity and cold	Non- and/or dominant	Waterproof	Thermic		
		Waterproof	Thermic	Waterproof *	
		Waterproof	Waterproof	Thermic	

# Glove sequences protecting against knife blows, use the anti-cut glove sequencing on the dominant hand; * If the worker feels the need to use this glove. ** Anti-cut gloves should also have thermal protection.

## Data Availability

The original contributions presented in the study are included in the article, further inquiries can be directed to the corresponding author.

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
