# Peer review of "Effect of Knife Use and Overlapping Gloves on Finger Temperature of Poultry Slaughterhouse Workers"

_ijerph, 2024, doi:10.3390/ijerph21101314_

Round 1

Reviewer 1 Report

Comments and Suggestions for Authors

1. Figure 1 represents the appearance of the gloves. The authors should provide in the article an additional table with detailed parameters of all gloves and their materials (glove size, glove weight, exact name of materials, polymer / fibrous composition of materials, thickness, type of material (rubber, knitwear and others), for knitwear – type of knitted weave, for all materials - porosity, bulk density, air permeability, hygroscopicity, water resistance). 

2. In the article, it is necessary to present the parameters of one type of knife, which was the same and constant for all employees (dimensions, weight, material of manufacture, shape and diameter of the knife at the point of grip by hand, the roughness coefficient of the knife surface in the area of contact with the palm of the hand).All these parameters significantly affect the operation and temperature of a person's hand. 

3. «It is noteworthy that 49.2% of workers who used a knife felt cold in their non-dominant hand.».   It is necessary to provide an explanation of the physiological causes of this effect. 

4. The authors should systematize the obtained quantitative patterns and present them in the form of graphs of the dependence of the skin temperature of the hands on two sides of the palm for the left and right hands depending on: 

- from the thickness of the insulating layer of gloves with the 1st glove; 

- from the thickness of the insulating layer of 2, 3 or more gloves together (taking into account their sequence on the hand);

- from the fibrous composition of gloves to a constant thickness;

- from porosity and breathability at constant thickness

and from other parameters of gloves and their materials listed in remark No. 1 (above). 

5. In conclusion, it is necessary to formulate specific recommendations, which gloves and in what sequence under what conditions should be worn with and without the use of a knife? What exactly is the temperature of the hand that such a method of using gloves with an external cooling of 12 degrees will provide? In the presented conclusion of the article, there are currently no final recommendations: what exactly and how should be done so that the temperature of the hands corresponds to physiological comfort?1. Figure 1 represents the appearance of the gloves. The authors should provide in the article an additional table with detailed parameters of all gloves and their materials (glove size, glove weight, exact name of materials, polymer / fibrous composition of materials, thickness, type of material (rubber, knitwear and others), for knitwear – type of knitted weave, for all materials - porosity, bulk density, air permeability, hygroscopicity, water resistance). 

2. In the article, it is necessary to present the parameters of one type of knife, which was the same and constant for all employees (dimensions, weight, material of manufacture, shape and diameter of the knife at the point of grip by hand, the roughness coefficient of the knife surface in the area of contact with the palm of the hand).All these parameters significantly affect the operation and temperature of a person's hand. 

3. «It is noteworthy that 49.2% of workers who used a knife felt cold in their non-dominant hand.».   It is necessary to provide an explanation of the physiological causes of this effect. 

4. The authors should systematize the obtained quantitative patterns and present them in the form of graphs of the dependence of the skin temperature of the hands on two sides of the palm for the left and right hands depending on: 

- from the thickness of the insulating layer of gloves with the 1st glove; 

- from the thickness of the insulating layer of 2, 3 or more gloves together (taking into account their sequence on the hand);

- from the fibrous composition of gloves to a constant thickness;

- from porosity and breathability at constant thickness

and from other parameters of gloves and their materials listed in remark No. 1 (above). 

5. In conclusion, it is necessary to formulate specific recommendations, which gloves and in what sequence under what conditions should be worn with and without the use of a knife? What exactly is the temperature of the hand that such a method of using gloves with an external cooling of 12 degrees will provide? In the presented conclusion of the article, there are currently no final recommendations: what exactly and how should be done so that the temperature of the hands corresponds to physiological comfort?

Author Response

Comments 1: 1. Figure 1 represents the appearance of the gloves. The authors should provide in the article an additional table with detailed parameters of all gloves and their materials (glove size, glove weight, exact name of materials, polymer / fibrous composition of materials, thickness, type of material (rubber, knitwear and others), for knitwear – type of knitted weave, for all materials - porosity, bulk density, air permeability, hygroscopicity, water resistance). 

Resposnse 1: The reviewer's request is pertinent for an experimental study or a laboratory study with strict control of variables. However, the present study is descriptive and was carried out in real work situations, analyzing the gloves that were used daily by workers from seven companies located in different regions of the country. Each company adopted gloves of different brands and models, and these were categorized in this study according to their general characteristics, cited in Figure 1. The researchers did not have access to more detailed information regarding the technical characteristics of each glove, nor did they have the possibility to perform specific tests, since data collection was limited to a single visit to each company. This information regarding the limitations of the present study was included in item “4.1. Strengths and limitations”.

Comments 2: 2. In the article, it is necessary to present the parameters of one type of knife, which was the same and constant for all employees (dimensions, weight, material of manufacture, shape and diameter of the knife at the point of grip by hand, the roughness coefficient of the knife surface in the area of contact with the palm of the hand).All these parameters significantly affect the operation and temperature of a person's hand. 

Response 2: We also consider that this request from the reviewer would be relevant for an experimental study or a laboratory study with strict control of variables. In the present descriptive study, conducted in real work situations in seven companies located in different regions of the country, it would be unfeasible to require all workers to use the same type of knife. Furthermore, in all slaughterhouses and meat processing industries, different types and sizes of knives are used according to the type of cut that is performed (different work activities). Therefore, since the present study did not aim to study the effects of using different types of knives, the requested information was not controlled.

Comments 3: 3. «It is noteworthy that 49.2% of workers who used a knife felt cold in their non-dominant hand.».   It is necessary to provide an explanation of the physiological causes of this effect. 

Response 3: The fact that 49.2% of workers who used a knife felt cold in their non-dominant hand may be directly related to contact with chilled pieces of meat (≤7 °C). While the dominant hand manipulates the knife, the non-dominant hand constantly manipulates the pieces of meat, with periods of intermittency that are probably insufficient for adequate physiological recovery. This issue is discussed in the paragraph between lines 346 and 353 of the article.

Comments 4: 4. The authors should systematize the obtained quantitative patterns and present them in the form of graphs of the dependence of the skin temperature of the hands on two sides of the palm for the left and right hands depending on: 

- from the thickness of the insulating layer of gloves with the 1st glove; 

- from the thickness of the insulating layer of 2, 3 or more gloves together (taking into account their sequence on the hand);

- from the fibrous composition of gloves to a constant thickness;

- from porosity and breathability at constant thickness

and from other parameters of gloves and their materials listed in remark No. 1 (above). 

Response 4: The suggestion made by the reviewer is interesting and would certainly result in a more robust article, however we do not have the data/information necessary to carry out the aforementioned analyses. Furthermore, a considerable number of tables/figures have already been presented in this article.

Comments 5: 5. In conclusion, it is necessary to formulate specific recommendations, which gloves and in what sequence under what conditions should be worn with and without the use of a knife? What exactly is the temperature of the hand that such a method of using gloves with an external cooling of 12 degrees will provide? In the presented conclusion of the article, there are currently no final recommendations: what exactly and how should be done so that the temperature of the hands corresponds to physiological comfort?

Response 5: As already highlighted in the previous answers, since this is a descriptive study, the conclusions are limited to observations of the realities found in the slaughterhouses analyzed. Based on the results of this study, specific recommendations were formulated regarding the type and sequence of gloves to be used in different work situations in slaughterhouses (Table 8). These recommendations aim to promote thermal protection for slaughterhouse workers who use or do not use knives.

Reviewer 2 Report

Comments and Suggestions for Authors

It is an overall well designed and presented research. The problem of the cold and repetitive motion induced discomfort and potential musculoskeletal issues is a significant one in occupational safety and health for the poultry workers. The advantages of this research include the diverse participating population and the various combination of the choice of the gloves. However, there are some minor issues that I suggest the authors to try to address or otherwise include in future research. First of all, it would be beneficial to measure the thermal insulation of the gloves that participants used, especially the multilayer combined thermal insulation. This can be done on a thermal hand manikin. Such data will provide quantitative comparisons of the different wearing conditions across different worksite, tasks, and glove conditions. Secondly, the validity of using thermal imaging on monitoring and recording hand skin temperature should be demonstrated with comparison to some wired sensors. This does not need to be extensive, just enough data to assess the consistency of the thermal imaging and variability compared to physiological sensors should be good. The article reads well, however, it would be better to have some editing, especially in the section for the description of the grouping of the participants/glove conditions. I found it a bit hard to follow the complex grouping conditions.

Comments on the Quality of English Language

Some professional editing will be beneficial but not necessary. 

Author Response

Comments 1: It is an overall well designed and presented research. The problem of the cold and repetitive motion induced discomfort and potential musculoskeletal issues is a significant one in occupational safety and health for the poultry workers. The advantages of this research include the diverse participating population and the various combination of the choice of the gloves. However, there are some minor issues that I suggest the authors to try to address or otherwise include in future research. First of all, it would be beneficial to measure the thermal insulation of the gloves that participants used, especially the multilayer combined thermal insulation. This can be done on a thermal hand manikin. Such data will provide quantitative comparisons of the different wearing conditions across different worksite, tasks, and glove conditions. Secondly, the validity of using thermal imaging on monitoring and recording hand skin temperature should be demonstrated with comparison to some wired sensors. This does not need to be extensive, just enough data to assess the consistency of the thermal imaging and variability compared to physiological sensors should be good. The article reads well, however, it would be better to have some editing, especially in the section for the description of the grouping of the participants/glove conditions. I found it a bit hard to follow the complex grouping conditions.

Response 1: The reviewer's considerations are very pertinent, and we are currently starting a new research project, in partnership with other laboratories at our university, with the aim of developing a thermal mannequin to test personal protective equipment used in slaughterhouses, including gloves.

As for the validity of using thermal images to monitor and record hand skin temperature, there are numerous published studies on this topic, some of which were cited in this study, with emphasis on a previous study published in IJERPH.

Regarding the quality of the English, the article was reviewed by a native-English speaking editor from a company specializing in proofreading technical texts (Proof attached.).
